# The Roles of GRETCHEN HAGEN3 (GH3)-Dependent Auxin Conjugation in the Regulation of Plant Development and Stress Adaptation

**DOI:** 10.3390/plants12244111

**Published:** 2023-12-08

**Authors:** Pan Luo, Ting-Ting Li, Wei-Ming Shi, Qi Ma, Dong-Wei Di

**Affiliations:** 1College of Life Science and Technology, Gansu Agricultural University, Lanzhou 730070, China; luopan@gsau.edu.cn; 2State Key Laboratory of Soil and Sustainable Agriculture, Institute of Soil Science, Chinese Academy of Sciences, Nanjing 210008, China; litingting@issas.ac.cn (T.-T.L.); wmshi@issas.ac.cn (W.-M.S.); 3University of Chinese Academy of Sciences, Beijing 100049, China

**Keywords:** auxin, IAA, GH3, chemical inhibitor, transcriptional regulation

## Abstract

The precise control of free auxin (indole-3-acetic acid, IAA) gradient, which is orchestrated by biosynthesis, conjugation, degradation, hydrolyzation, and transport, is critical for all aspects of plant growth and development. Of these, the GRETCHEN HAGEN 3 (GH3) acyl acid amido synthetase family, pivotal in conjugating IAA with amino acids, has garnered significant interest. Recent advances in understanding GH3-dependent IAA conjugation have positioned GH3 functional elucidation as a hot topic of research. This review aims to consolidate and discuss recent findings on (i) the enzymatic mechanisms driving GH3 activity, (ii) the influence of chemical inhibitor on GH3 function, and (iii) the transcriptional regulation of GH3 and its impact on plant development and stress response. Additionally, we explore the distinct biological functions attributed to IAA-amino acid conjugates.

## 1. Introduction

Auxins, a group of phytohormones, are integral to the regulation of plant development and stress responses [1,2]. Previous research has identified three naturally occurring auxins: indole-3-acetic acid (IAA), phenylacetic acid (PAA), and 4-chloro-indole-3-acetic acid (4-Cl-IAA), with IAA being the most prevalent and significant in plants [1]. The meticulous modulation of IAA levels, governed by biosynthesis, transport, and inactivation, is essential for normal plant growth, development, and adaptation to both biotic and abiotic environmental stresses [1,3,4].

Four primary pathways have been documented for IAA inactivation in plants: (i) IAA CARBOXYL METHYLTRANSFERASE1 (IAMT1) converts IAA to methyl IAA; (ii) UDP-glucosyltransferase (UGTs) generate ester-linked IAA conjugates; (iii) GRETCHEN HAGEN 3 (GH3) acyl amido synthetases facilitate the formation of amide-linked IAA conjugates; and (iv) IAA oxidation carried out by DIOXYGENASE FOR AUXIN OXIDATION (DAO) [5,6,7,8,9,10]. Methyl IAA and ester-linked IAA, both subject to reconversion into IAA by specific hydrolases, are predominantly regarded as forms of IAA storage [11]. Contrastingly, the reversibility of amide-linked conjugates varies based on the amino acid involved. Most, such as IAA-alanine (IAA-Ala), IAA-phenylalanine (IAA-Phe), and IAA-leucine (IAA-Leu), revert to free IAA, while others like IAA-glutamate (IAA-Glu) and IAA-aspartate (IAA-Asp) are directly degraded [5,11,12,13]. However, recent findings suggest that IAA-Glu and IAA-Asp are also hydrolyzed to free IAA by IAA-LEU-RESISTANT1 (ILR1), and that DAO-dependent oxidation also occurs by oxidizing IAA-Asp and IAA-Glu to oxIAA-Asp and oxIAA-Glu, which in turn are hydrolyzed to oxIAA in the presence of the hydrolase ILR1, rather than by direct IAA oxidation to oxIAA. This suggests that GH3-dependent IAA conjugation may be a key node in IAA storage and IAA oxidative degradation [6,14]. 

The first GH3 gene was identified from *Glycine max* as a rapid early auxin-responsive gene. Subsequent research has established the widespread distribution of the GH3 gene family across the plant kingdom, encompassing species from *Arabidopsis thaliana* to *Oryza sativa*, *Zea mays*, *Triticum aestivum*, and even non-vascular plants like *Physcomitrella patens* and *Marchantia polymorpha* [8,11,15,16,17,18,19]. Based on sequence homology and substrate specificity, the GH3 family in Arabidopsis is categorized into three distinct groups: Group I, II, and III. Group I GH3 genes are known to encode enzymes that synthesize amides from jasmonic acid (JA) or salicylic acid (SA). Group II GH3 enzymes function as IAA-amido synthetases, and Group III has been shown to catalyze the conjugation between amino acids and 4-substituted benzoates or indole-3-butyric acid (IBA) [8,20]. Emerging evidence underscores the role of GH3-mediated IAA conjugation not only in modulating free IAA availability but also in its potential as a signaling molecule or inhibitor, impacting plant growth and development [21,22,23,24]. This review will delve into the biochemical mechanisms of GH3-mediated IAA conjugation and its transcriptional regulation.

## 2. The Catalytic Mechanisms and Substrate Specificity of GH3 Acyl Acid Amido Synthetase Enzyme

Chen et al. firstly utilized a combination of initial velocity and product inhibition analyses, alongside mass spectrometry, to delineate the kinetic and chemical mechanisms governing OsGH3.8 activity [25]. They discovered that the conjugation of IAA with Asp operates via a ‘Bi Uni Uni Bi Ping Pong’ mechanism, as depicted in Figure 1A. The process initiates with the binding of IAA and ATP, in the presence of Mg^2+^, to the unoccupied enzyme. This interaction results in the formation of an adenylated IAA intermediate (IAA-AMP) and the concurrent release of pyrophosphate (PPi). Following this, Asp attaches to the enzyme•IAA•AMP complex, leading to the displacement of AMP and the establishment of an amide linkage between IAA and Asp. The final reaction products, IAA-Asp and AMP, are then released from the OsGH3.8 enzyme’s active site [25]. 

Structural analyses of the GH3 enzyme in Arabidopsis, grape, and rice have illuminated that both monocotyledons and dicotyledons employ a similar mechanism for AMP and IAA binding [26]. The GH3 enzyme exhibits distinct acyl acid binding preferences, with specific residues within its active site conferring selectivity for particular substrates [26]. In OsGH3.8, the amino acids Arg130 and Leu137 play a crucial role in substrate specificity. The mutation of Arg130 to Leu (Arg130-Leu) shifts the enzyme’s substrate preference from IAA to benzoate/SA, while an Arg130-Thr substitution favors JA over IAA. Similarly, Leu137-Ser mutation leads to a benzoate/SA preference, and the replacement of Leu137 with Arg/Ile induces a preference for JA [25,26]. GH3 proteins also exhibit amino acid specificity; for instance, the carboxylate group of Asp is a determinant for the active site’s specificity in OsGH3.8 [25,26]. Ser341 participates in adenylate formation by forming hydrogen bonds with phosphate groups [26]. Moreover, Mg^2+^ is essential for the enzyme’s maximum activity, aiding in AMP orientation, with Glu342 being critical for Mg^2+^ coordination [25,26,27]. Additionally, seven residues identified through the sequence comparison of acyl-substrate binding sites (Arg130, Leu137, Valine 174 (Val174), Leu175, Methionine 337 (Met337), Alanine 339 (Ala339), and Tyrosine 344 (Tyr344)) are thought to be involved in the substrate-specific selection of IAA [26].

It has been observed that the residues involved in the AMP binding site of acyl adenylate cleavage enzymes exhibit a high degree of conservation across the GH3 protein family. In contrast, the residues that interact with acyl substrates show variability, accommodating the binding of diverse substrates [25,26,28,29]. This variation in amino acid residues leads to the formation of different binding pockets, each tailored for specific substrates [29]. This may account for the fact that auxin function at all stages of plant growth and development, a process that offers great flexibility in regulating auxin action is necessary. The existence of a complex auxin conjugation system, as evidenced by these variations in GH3 proteins, is likely a strategic evolutionary development to facilitate this flexibility [28].

## 3. The Modulation of IAA Homeostasis by Small Chemical Molecules via the Inhibition of GH3 Enzyme Activity

The functional redundancy of class II GH3 enzymes in plants presents a challenge to traditional genetic approaches when exploring their biological roles [30]. To overcome this obstacle, small molecule inhibitors have emerged as a powerful alternative. These inhibitors can be applied to any plant tissue at any developmental stage, given in appropriate concentrations. Their utility lies in the ability to bypass gene redundancy and the potential detrimental effects of lethal mutations often associated with simultaneous multi-gene mutations [31]. To date, three potent inhibitors have been identified that modulate GH3-dependent IAA conjugation: adenosine-59-[2-(1H-indol-3-yl) ethyl] phosphate (AIEP), kakeimide (KKI), and N-[4-[[6-(1H-pyrazol-1-yl)-3-pyridazinyl]amino] phenyl]-3-(trifluoromethyl) benzamide (nalacin), as represented in Figure 1B [30,31,32].

### 3.1. AIEP, the First Chemical Inhibitor of Auxin Conjugation

The GH3-dependent IAA conjugation initiates when the unbound GH3 enzyme interacts with ATP and IAA, leading to the formation of IAA•AMP. Böttcher et al. engineered and synthesized a stable analogue of IAA•AMP, named AIEP. This molecule competes with ATP and IAA for the binding sites on the GH3 enzyme at the onset of catalysis [30]. The competitive inhibitory effect of AIEP on ATP and IAA binding was validated through substrate velocity experiments involving VvGH3.1 and VvGH3.6 from grape. However, the study did not extend to phenotypic examinations to assess the broader biological impacts of this inhibition (Figure 1). 

### 3.2. KKI, a Specific Inhibitor of IAA-Conjugating GH3 Enzymes

In the quest to identify inhibitors of IAA-conjugating GH3 enzymes, researchers leveraged Arabidopsis *AtGH3.6* overexpression plants as a biological assay system. They screened a synthetic chemical library comprising 10,000 compounds for agents capable of reverting the altered root hair growth phenotype of AtGH3.6-overexpressed lines. This led to the initial identification of compound ‘1’, followed by the synthesis of 25 derivatives of ‘1’. Among these, kakeimide (KKI) emerged as a highly potent inhibitor. (Figure 1B). KKI functions by directly interacting with the IAA binding site within the GH3•ATP complex, forming a stable GH3•ATP•KKI ternary complex that impedes the synthesis of IAA-amino acid conjugates [32]. Validation experiments demonstrated KKI’s effectiveness in targeting the IAA binding sites of various GH3 enzymes, notably VvGH3.1, AtGH3.5, and OsGH3.8, while sparing the IBA binding site of AtGH3.15. This specificity, coupled with KKI’s lack of interference in jasmonic acid (JA) homeostasis, underscores its role as a specific inhibitor of IAA-conjugating GH3 enzymes [32].

### 3.3. Nalacin, a Potent Inhibitor Targeting Group II GH3 Enzymes

Nalacin was identified from a chemical screen by observing the auxin-related root phenotypes in the Arabidopsis wild-type Col-0 (Figure 1B). Subsequent studies have shown that nalacin competitively inhibits substrate acceptance by AtGH3.6 and AtGH3.11 through trifluoromethyl phenyl occupancy of the IAA binding site of AtGH3s, suggesting that nalacin also functions in the first step of the ‘Bi Uni Uni Bi Ping Pong’ reaction of GH3 enzymes. Unlike KKI, which selectively inhibits only class II GH3 enzymes, nalacin also impedes the formation of JA amino acid conjugates mediated by AtGH3.11, albeit through a distinct binding mode [31]. Consequently, there is potential for further chemical modifications of nalacin to enhance its selective inhibition of different GH3 enzyme members.

## 4. The Transcriptional Control of GH3 Enzymes in Plant Growth, Development, and Stress Adaptation

Research into GH3-dependent IAA conjugation has revealed its critical role in the modulation of plant growth, development, and stress responses across a variety of species, with extensive studies conducted particularly in maize, wheat, rice, and Arabidopsis. The forthcoming sections will focus on the advances in understanding GH3-dependent IAA conjugation in Arabidopsis and rice. Additionally, progress in other species, reflecting the broader impact and relevance of GH3 enzymes in plant biology, has been systematically compiled in Table 1.

### 4.1. GH3-Dependent IAA Conjugation Is Involved in Regulating Multiple Developmental Processes

In Arabidopsis, eight Group II *GH3* genes are involved in catalyzing IAA conjugation to amino acids. Due to redundant gene functions, mutations in single genes result in only subtle phenotypic changes and modified sensitivity to exogenous IAA [54]. In contrast, mutants with overexpressed *GH3* genes, obtained through activation tag insertion, provide a more discernible phenotype for study. For instance, *AtGH3.2* and *AtGH3.6* were identified through the screening of their overexpressed mutants, *ydk1-D* and *dfl1-D*, respectively [54,55]. Interestingly, however, despite all overexpressing genes being closely related to *GH3* Group II family members, they still showed inconsistent phenotypes. Under various light conditions, the *dfl1-D* mutant displayed shortened hypocotyls exclusively in light, while the *ydk1-D* mutant showed this phenotype under both light and dark conditions. Additionally, the *ydk1-D* mutant had a shorter primary root but did not exhibit significant difference in susceptibility to auxin-mediated root growth inhibition. In contrast, the *dfl1-D* mutant was resistant to IAA-mediated root growth inhibition and did not present a short-root phenotype compared with the wild type [54,55]. Furthermore, several *Group II GH3* genes are transcriptionally induced by IAA, whereas *AtGH3.9* is repressed in response to exogenous IAA application [56]. Altogether, these data indicate that, while Group II GH3 members share some commonalities and function in a similar pathway by regulating free IAA conjugation, they each play distinct roles in plant development, which may be due to differences in their tissue specificity and/or hormonal regulation (e.g., IAA, JA, etc.). 

As early auxin response genes, Group II *GH3* genes play a significant role downstream of auxin response factors (ARFs), which are key elements in the auxin signaling pathway [57]. Research has elucidated the involvement of ARF6, AtARF7, AtARF8, and AtARF17 in the transcriptional regulation of several *AtGH3* genes. AtARF7 and AtARF8 are known to positively regulate the transcription of *AtGH3.2*, *AtGH3.5*, and *AtGH3.6*, influencing hypocotyl elongation under different light conditions [54,58]. In contrast, AtARF17 has a negative regulatory role on *AtGH3.5* and *AtGH3.6* transcription, which is essential for proper plant development. The microRNA AtmiR160 directly targets *AtARF17* mRNA, which is crucial for normal leaf and root growth [59]. In terms of adventitious root development, AtARF6 and AtARF8 act as positive regulators, while AtARF17 functions as a negative regulator. They co-regulate the transcription of *AtGH3.3*, *AtGH3.5*, and *AtGH3.6*, impacting JA conjugation with amino acids, but not IAA conjugation [60]. Additionally, AtARF7 actives another auxin-induced transcription factor (TF), AtMYB77, which subsequently upregulates *AtGH3.2* and *AtGH3.3* to control root development [61]. AtMYB30, another MYB TF, directly binds to the promoters of *AtGH3.2* and *AtGH3.3*, repressing their transcription to foster root elongation [62]. WRINKLED1 (WRI1), yet another TF, binds to the *AtGH3.3* promoter. Although *AtGH3.3* transcription increases in the *Atwri1-1* mutant, AtWRI1 does not appear to repress *AtGH3.3* directly, suggesting it may function as a transcriptional co-factor [63,64]. Subsequent research identified that AtTCP20, a TF that interacts with AtWRI1, binds to the *AtGH3.3* promoter, and activates its transcription [63]. Electrophoretic mobility shift assays have shown that the addition of AtWRI1 decreases the binding activity of AtTCP20 to the *AtGH3.3* promoter in a dose-dependent manner, indicating that AtWRI1 regulates *AtGH3.3* transcription by antagonizing AtTCP20’s binding [63]. Several basic leucine zipper (bZIP) transcription factors, including AtbZIP2, 11, 44, 53, and 63, also directly bind to the *AtGH3.3* promoter and enhance its transcription [65]. Furthermore, cytokinin influences the root meristem by maintaining IAA concentration, with AtARR1 directly activating *AtGH3.17* transcription [66]. There are also indications of TFs AtHLS1, AtURO, and AtSTY1 being involved in modulating GH3-dependent IAA conjugation in plant developmental processes, although direct regulatory evidence is yet to be established (Figure 2) [67,68,69].

In rice, the Group II GH3 family, which includes *OsGH3.1*, *OsGH3.2*, *OsGH3.5*, *OsGH3.8*, and *OsGH3.13*, plays a significant role in various developmental aspects such as shoot height, leaf angle, floret fertility, and tiller number. These effects are primarily mediated through the modulation of IAA conjugation [70,71,72,73,74]. Recent findings highlight several TFs that directly activate the transcription of *OsGH3* genes, leading to decreased levels of free IAA and consequent impacts on rice morphology. For example, OsbZIP49 has been shown to upregulate *OsGH3.2* and *OsGH3.13* by binding to TGACG motifs in their promoters. This upregulation results in reduced free IAA concentrations, affecting cell elongation mediated by expansins and thus impacting shoot gravitropism [75]. Furthermore, *OsGH3.2* is regulated by OsARF8, a downstream target of OsmiR167, indicating a critical role for the OsmiR167-OsARF8-OsGH3.2 pathway in cellular auxin homeostasis, particularly in response to exogenous auxin [76]. Another ARF, OsARF19, has been found to reduce free IAA levels by activating *OsGH3.5*, thereby influencing leaf angulation. Intriguingly, the transcription of *OsARF19* itself is induced by IAA and brassinolide (BR), suggesting the OsARF19-OsGH3.5 module’s involvement in integrating IAA-BR signals [73]. Additionally, OsSPL7, a target of OsmiR156f, directly activates *OsGH3.8*, affecting tiller number and shoot height [70]. The involvement of OsMADS1 and OsMADS6 in binding to the *OsGH3.8* promoter and regulating floret fertility has also been documented [77]. Collectively, these findings demonstrate that *Group II GH3* genes, together with their upstream TFs, form a complex regulatory network that integrates light, miRNA, and hormonal signals to control plant growth and development (Figure 3) [78].

### 4.2. The Integration of Hormonal Signals in GH3-Dependent IAA Conjugation’s Responses to Abiotic Stresses

#### 4.2.1. Drought Stress

In Arabidopsis, *AtGH3.5* has been observed to respond rapidly to drought conditions, with the *wes1-D* mutant, which overexpresses *AtGH3.5*, exhibiting enhanced drought resistance [79]. A subsequent study revealed that AtMYB96 modulates the expression of several *AtGH3* genes, including *AtGH3.3*, *AtGH3.5*, and *AtGH3.6*, under drought stress through an abscisic acid (ABA)-dependent pathway [80]. This finding underscores the importance of ABA signaling in modulating *GH3* gene expression during drought response. Additionally, the *gh3oct* mutant, with knockouts of all *Group II GH3* genes (*GH3.1*,*2*,*3*,*4*,*5*,*6*,*9*,*17*), exhibits increased drought tolerance, further highlighting the role of these genes in drought response mechanisms [81]. In rice, the activation of *OsGH3.13* has been linked to a reduction in free IAA levels, leading to a structural adaptation in the leaves, such as thicker blades, which enhance drought tolerance by minimizing water loss [74]. However, the response to drought stress in rice is complex, as evidenced by the contrasting effects observed with *OsGH3.2*. While *OsGH3.2* is also upregulated in response to drought, its overexpression leads to decreased drought tolerance. This discrepancy is attributed to the inhibition of carotenoid biosynthesis by overexpressed *OsGH3.2*, which consequently reduces ABA synthesis. This is in stark contrast to the increased ABA levels seen in lines overexpressing *OsGH3.13* [72]. These findings collectively indicate that, while *GH3* genes are integral to stress responses through IAA homeostasis regulation, the distinct spatial–temporal expression patterns and secondary growth effects can result in varying stress sensitivities [72,74]. 

#### 4.2.2. Temperature (Heat/Cold/Freezing) Stress

In Arabidopsis, the transcription of *AtGH3.5* is notably responsive to temperature extremes, showing increased levels under both low (4 °C) and high (37 °C) temperature conditions. The *wes1-D* mutant, characterized by the overexpression of *AtGH3.5*, shows increased survival after exposure to freezing temperatures (−7 °C). This suggests a broad regulatory role for *AtGH3.5* across a spectrum of temperature stresses [82]. In rice, the overexpression of *OsGH3.2* leads to a reduction in free IAA levels, thereby activating cold-responsive genes and enhancing the plant’s ability to scavenge reactive oxygen species (ROS). Consequently, this confers increased resistance to cold stress [72]. 

#### 4.2.3. Salt and Osmotic Stress

All root-expressed Group II *GH3* genes in Arabidopsis are upregulated following treatment with NaCl at concentrations of 75 mM and 150 mM. The *Atgh3oct* mutant, with combined knockouts of all Group II *GH3* genes, exhibits greater resilience to NaCl stress compared to the wild type [81]. This enhanced tolerance also extends to sorbitol and mannitol exposure, suggesting that Group II GH3s may confer broad osmotic stress resistance, inclusive of salinity stress. Further investigation reveals that NaCl treatment increases *AtACS2* transcription, leading to the accumulation of the ethylene precursor ACC, which in turn downregulates *AtGH3.5* and *AtGH3.9* transcription, maintaining free IAA levels and primary root growth [83]. Additionally, both osmotic and salt stresses are known to stimulate the accumulation of ABA, which reduces free IAA content, inhibiting lateral root (LR) development. At the same time, these stresses activate AtWRKY46, a transcription factor that suppresses *AtGH3.1* transcription, thus maintaining free IAA levels and LR development. Additionally, ABA inhibits *AtWRKY46* transcription. This antagonistic interaction between ABA and AtWRKY46 fine-tunes free IAA levels and LR development, allowing for improved adaptation to osmotic and salt stresses [84]. 

#### 4.2.4. Ammonium (NH_4_^+^) Stress

NH_4_^+^ serves as a vital nitrogen source for plants, but when available in excess, it can be detrimental to growth [85,86]. Prior research indicates that high NH_4_^+^ levels lead to a reduction in free IAA [87,88,89]. In Arabidopsis, elevated NH_4_^+^ conditions trigger the induction of nearly all Group II *GH3* genes, which in turn accelerates the conjugation of free IAA to amino acids [87]. Concurrently, high NH_4_^+^ levels also enhance the transcription of *AtWRKY46*, a transcription factor that binds to the promoters of *AtGH3.1* and *AtGH3.6*, repressing their transcription. This response serves to maintain free IAA levels and support primary root growth under high NH_4_^+^stress conditions. The overexpression of *AtWRKY46* improves NH_4_^+^ tolerance, suggesting its critical role in modulating primary root development during NH_4_^+^ stress [90]. 

#### 4.2.5. Pathogen Stress

In response to pathogen attacks, plants activate a comprehensive defense strategy: (1) they initiate a hypersensitive response leading to rapid programmed cell death at the infection site alongside other defense responses; (2) they activate systemic-acquired resistance (SAR) in distal tissues; and (3) they activate basal immunity to limit pathogen growth [91]. GH3-dependent IAA conjugation is intricately involved in these plant defense mechanisms. In Arabidopsis, the pathogens *B. cinerea* and *P. syringae pv tomato* (*Pst*) DC3000 significantly upregulate *AtGH3.2* and *AtGH3.3* transcription. Loss-of-function mutations in *AtGH3.2* or *AtGH3.4* enhance resistance to both pathogens, suggesting that *AtGH3.2*, *AtGH3.3*, and *AtGH3.4* may negatively influence the plant’s response to *B. cinerea* and *Pst* DC3000 [92]. The overexpression of mutant *gh3.5-1D* shows a compromised hypersensitive response but retains normal SAR and basal immunity, whereas the *Atgh3.5* mutant exhibits a defective SAR response yet maintains a typical hypersensitive response and basal immunity. In contrast, the *dfl1-D* mutant displays altered hypersensitive and basal immune responses [91]. 

In rice, IAA has been linked to increased susceptibility to various pathogens, partly due to IAA-induced expansins that relax the cell wall, a plant’s primary defense barrier [93]. Overexpressing certain Group II *GH3* genes, such as *OsGH3.1*, *OsGH3.2*, and *OsGH3.8*, bolsters resistance against bacterial and fungal pathogens. This resistance is attributed to the suppression of pathogen-induced IAA accumulation, leading to the downregulation of expansin expression and subsequent stabilization of the cell wall [71,93,94]. 

The varied effects of Group II *GH3* genes on pathogen stress between rice and Arabidopsis could be attributed to multiple factors. In rice, OsGH3s appear to primarily contribute to forming a robust cell wall barrier, which acts as the first line of defense against pathogen entry, without further invoking a hypersensitive response or SAR [39,94]. In contrast, the expression of *AtGH3* in Arabidopsis is highly specific to tissue type and developmental stage, and unspecific overexpression could lead to unintended consequences, such as disruptions in the levels of IAA and salicylic acid (SA), which are critical for the plant’s defense response [91,92]. Additionally, the GH3-catalyzed IAA-Asp conjugate is hypothesized to act as a susceptibility signal [23]. While GH3 activity reduces free IAA levels, it simultaneously results in the accumulation of IAA-Asp. The specific distribution and balance of IAA and IAA-Asp within the plant may significantly influence the outcome of GH3’s role in pathogen response [92].

## 5. Atypical Roles of Group II GH3 and IAA-Amino Acids

The exploration of Group II GH3 enzymes as IAA-acyl acid amino synthetases in 2002 marked a significant advancement in our understanding of these enzymes [11]. In addition to the determination of their three-dimensional structures and the key amino acid sites for enzyme activity, research has expanded their known functions to include the conjugation of other auxins like PAA and IBA with amino acids, IAA conjugation with proteins, and the metabolism of auxinic herbicides [95]. Furthermore, AtGH3.15, previously an undefined member of Group III, is now recognized as an IBA-conjugating acyl acid amido synthetase. IBA serves as a precursor to IAA, which is converted through the peroxisomal β-oxidation pathway [96]. Intriguingly, AtGH3.13, AtGH3.14, and AtGH3.16—other members of Group III—possess acyl acid binding sites similar to AtGH3.15, suggesting their possible involvement in IAA homeostasis [20,95]. Here, we will discuss the atypical roles of Group II GH3 and IAA-amino acids. 

### 5.1. The Roles of Group II GH3 beyond the Catalyzation of IAA-Amino Acid Conjugate Formation

The capacity of Group II GH3 enzymes extends beyond the synthesis of IAA-amino acid conjugates. Research using recombinant GH3 IAA-amino acid synthetase from pea has revealed the enzyme’s ability to conjugate IAA not only to aspartate but also to proteins in immature seeds. The proposition that IAA conjugation to proteins may serve a regulatory function acting as a prosthetic group and influencing protein activity via posttranslational modifications is a compelling avenue for further exploration [97,98]. Besides proteins, Group II GH3 enzymes also facilitate the conjugation of PAA to amino acids. PAA, another natural but less active auxin than IAA, exhibits unique distribution and transport characteristics, implying a role in sustaining the auxin equilibrium necessary for plant cellular processes [99,100,101]. Notably, PAA can stimulate GH3-dependent IAA conjugation, while high IAA levels can suppress PAA biosynthesis, underscoring the importance of Group II GH3 enzymes in balancing IAA/PAA ratios [99]. Moreover, this enzyme group, along with AtGH3.17, has been implicated in the detoxification of the auxinic herbicide 2,4-DB, suggesting their potential application in herbicide resistance strategies [102].

### 5.2. The Specialized Functions of IAA-Amino Acid Conjugates beyond Their Role as Auxin Stock

Previous studies found that the exogenous addition of IAA-aa and IAA both rapidly increased content-free IAA levels and exhibited similar high growth factor phenotypes, suggesting that IAA-aa is a storage form of IAA [103]. Subsequent studies found that these IAA-aa, IAA-Leu, and IAA-Ala could be reversibly converted to free IAA by the action of the hydrolases, IAA-LEUCINE RESIS TANT1 (ILR1), ILR1-LIKE proteins (ILLs), and IAA-ALANINE RESISTANT3 (IAR3) [5,104,105]. IAA-Glu and IAA-Asp, once considered only as degradation intermediates, are now recognized as reversible storage forms [14]. Beyond storage, IAA-aa have been identified as possessing unique biological functions. IAA-Trp, for instance, acts as a ‘super inactivator’ by not only consuming free IAA for its synthesis but also antagonizing the activity of residual IAA, with IAA-Trp significantly mitigating root inhibition effects caused by IAA [24]. IAA-Asp has been reported to have more diverse roles: (1) correcting the temperature sensitivity of henbane (*Hyoscyamus muticus*) XIlB2 (temperature-sensitive variant) cells [22]; (2) IAA-Asp directly and specifically enhance the pea (*Pisum sativum*) responses to abiotic stress by increasing the antioxidant enzyme activity and then reducing the H_2_O_2_ concentration [23]; (3) IAA-Asp as a ripening signal in grapes (*Vitis vinifera*) can be perceived at a certain stage of fruit development; however, the mechanism of sensing remains unknown [21]; and (4) IAA-Asp promotes pathogen development in plants by regulating the transcription of virulence genes [92]. These insights suggest that IAA-amino acids are not just byproducts of GH3 activity but are biologically active molecules with specific roles. 

## 6. Concluding Remarks

Our review underscores that Group II GH3 enzymes and IAA-amino acids are integral components of plant biology, with roles extending beyond the simple conversion of free IAA to its stored form. The functional diversity of these enzymes and their products in plant growth, development, and stress adaptation is a testament to the complexity of plant hormonal regulation. Each *GH3* gene exhibits specific spatial and temporal expression patterns, contributing to a nuanced regulatory network. These findings challenge the traditional view of IAA-amino acids as mere storage forms and highlight the need to consider their biological activity in understanding plant physiology and development.

## Figures and Tables

**Figure 1 plants-12-04111-f001:**
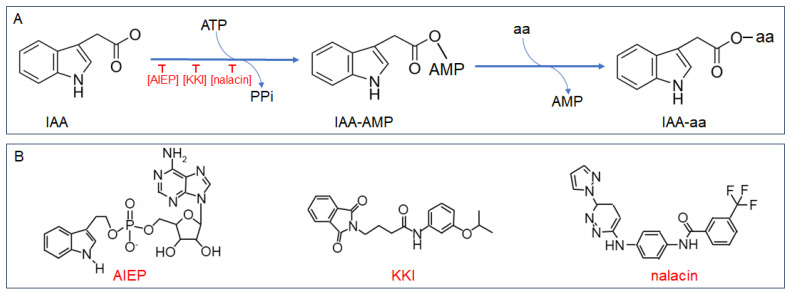
Catalytic reaction and inhibitor structures of Group II GH3 amido synthetase. (**A**) Schematic representation of the total reaction mediated by Group II GH3 amido synthetases. (**B**) Chemical structures of inhibitors targeting Group II GH3 amido synthetases: AIEP (adenosine-59-[2-(1H-indol-3-yl) ethyl] phosphate), KKI (kakeimide), and nalacin (N-[4-[[6-(1H-pyrazol-1-yl)-3-pyridazinyl] amino] phenyl]-3-(trifluoromethyl)benzamide).

**Figure 2 plants-12-04111-f002:**
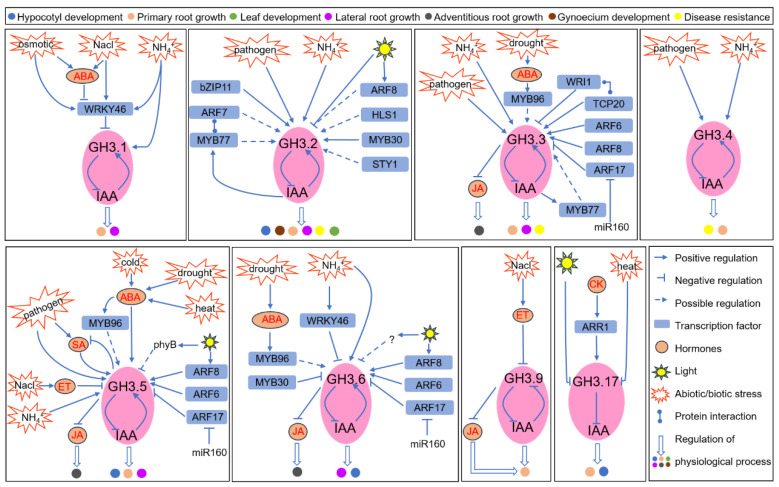
Regulatory network of Arabidopsis GH3-dependent IAA conjugation in abiotic and biotic stress responses. Abbreviation: NH_4_^+^, ammonium; WRKY46, WRKY DNA-BINDING PROTEIN46; bZIP11, basic leucine Zipper11; ARF6/7/8/17, auxin response factor6/7/8/17; MYB30/77/96, myeloblastosis30/77/96; WRI1, WRINKLED1; TCP20, TEOSINTE BRANCHED 1; CYCLOIDEA, PCF (TCP)-DOMAIN FAMILY PROTEIN 20; ARR1, Arabidopsis response regulator1; miR160, microRNA; IAA, indole-3-acetic acid; ABA, abscisic acid; JA, jasmonic acid; ET, ethylene; SA, salicylic acid; CK, cytokinin.

**Figure 3 plants-12-04111-f003:**
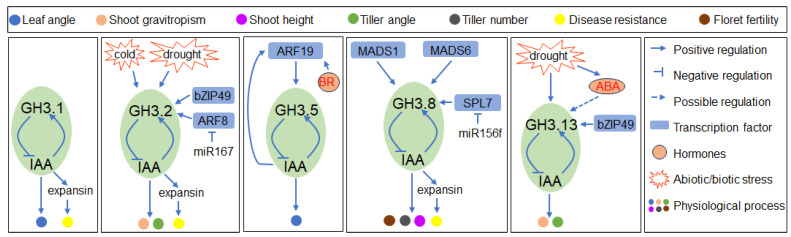
Role of rice GH3-dependent IAA conjugation in response to abiotic and biotic stresses and its transcriptional regulation. Abbreviation: bZIP49, basic leucine Zipper49; ARF8/19, auxin response factor; MADS1/6, MADS-domain transcription factor1/6; SPL7, SQUAMOSA PROMOTER BINDING PROTEIN-LIKE 7; miR156f/167, microRNA156f/167; IAA, indole-3-acetic acidl; BR, brassinolide; ABA, abscisic acid.

**Table 1 plants-12-04111-t001:** Compilation of Group II GH3 genes across various species and their identified biological roles.

Species	Members	TFs	Biological Process	Ref.
*Physcomitrella patens*	*PpGH3.1*		High temperature and salt tolerance	[33,34]
	*PpGH3.2*			
*Stylosanthes guianensis*	*SgGH3.1*		Chilling and cold tolerance	[35]
*Dianthus caryophyllus*	*DcGH3.1*		Adventitious root development	[36]
*Brassica oleracea*	*BoGH3.12*		Cadmium tolerance	[37]
*Pisum sativum*	*PsGH3.5*		Seedlings development	[38]
*Cucumis sativus*	*CsGH3.5*		Adventitious root formation	[39]
*Capsicum chinense*	*CcGH3*		Fruit ripening	[40]
*Solanum lycopersicum*	*SlGH3.8*	YABBY2b	Plant height	[41]
	*SlGH3.2*		Fruit ripening	[42]
	*SlGH3.15*		Lateral root development; gravitropism	[43]
*Coffea canephora*	*CcGH3.1*		Somatic embryogenesis	[44,45]
	*CcGH3.6*			
	*CcGH3.17*			
*Zea mays*	*ZmGH3.2*	DREB2A	Seed aging tolerance	[46]
*Vitis vinifera*	*VvGH3.1*		Berry ripening	[30]
	*VvGH3.2*			[21]
	*VvGH3.6*		Tissue auxin homeostasis	[28,30]
*Citrus sinensis*	*CsGH3.1*		Susceptibility to pathogen	[47]
	*CsGH3.1L*			
*Malus sieversii*	*MsGH3.5*	RR1a	Shoot and root development	[48]
*Malus domestica*	*MdGH3-2*	bHLH3	Leaf shape	[49]
*Castanea sativa*	*CsGH3.1*		Adventitious root development	[50]
*Carya cathayensis*	*CcGH3*		Grafting	[51]
*Picea abies*	*PaGH3.gII.8*		Tissue auxin homeostasis	[52]
	*PaGH3.gII.9*			
	*PaGH3.17*			
*Betula platyphylla*	*BpGH3.3*		Tissue auxin homeostasis	[53]
	*BpGH3.5a*			
	*BpGH3.5b*			
	*BpGH3.9*			

## Data Availability

Not applicable.

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
