# Peer review of "The Roles of GRETCHEN HAGEN3 (GH3)-Dependent Auxin Conjugation in the Regulation of Plant Development and Stress Adaptation"

_plants, 2023, doi:10.3390/plants12244111_

Round 1

Reviewer 1 Report

Comments and Suggestions for Authors

Review Report of the review article "Roles of GRETCHEN HAGEN3 (GH3)-dependent auxin Conjugation in the Regulation of Plant Development and Stress Adaptation.

The review MS provides a substantial advancement and consolidate discussion on recent findings on (i) the enzymatic mechanisms underpinning GH3 activity, (ii) the role of chemical inhibitors in the modulation of GH3 function, and (iii) the transcriptional regulation of GH3 and its impact on plant development and stress response. Authors claimed that, the GRETCHEN HAGEN 3 (GH3) acyl acid amido synthetase family has garnered significant interest as is critical for all aspects of plant growth and development.

The manuscript is well-written, and the information they introduced seems sound and practically useful and convincing.

They have collected most recent articles published under topics and reproduced the summery in a well manners. How the GH3 genes are synthesized and regulated in plants specially in Arabidopsis and rice, they have demonstrated clearly. Also included their modulation and inhibition for active function in diverse enzymatic system. Further, described the regulation of GH3 Enzymes in Plant Growth, Development, and Stress Adaptation.

Also summarized functions in biological pathways which is very important and necessary for further research

In conclusion, this review marks a substantial stride in comprehending the genetic and molecular underpinnings of GH3 genes and their stress responsive functions in plants which might be useful for further plant stress resistance research and crop improvement.

Minor comments:

L179: adventitious root (AR) came only one time. So, no need AR

Similarly, Electrophoretic mobility shift assays (EMSA), and cytokinin (CK) and other if any

Some genes are not italicized. Check throughout the text

L311-320: need some references if available

Genes (transcription factors) names should be italic.

Some references were old, revised it accordingly where necessary.

Some grammatical mistakes are present in the MS, revise the MS from the English speaker in the revised version.

Comments on the Quality of English Language

Minor editing of English language required

Author Response

Reviewer 1:

The review MS provides a substantial advancement and consolidate discussion on recent findings on (i) the enzymatic mechanisms underpinning GH3 activity, (ii) the role of chemical inhibitors in the modulation of GH3 function, and (iii) the transcriptional regulation of GH3 and its impact on plant development and stress response. Authors claimed that, the GRETCHEN HAGEN 3 (GH3) acyl acid amido synthetase family has garnered significant interest as is critical for all aspects of plant growth and development.

The manuscript is well-written, and the information they introduced seems sound and practically useful and convincing.

They have collected most recent articles published under topics and reproduced the summery in a well manners. How the GH3 genes are synthesized and regulated in plants specially in Arabidopsis and rice, they have demonstrated clearly. Also included their modulation and inhibition for active function in diverse enzymatic system. Further, described the regulation of GH3 Enzymes in Plant Growth, Development, and Stress Adaptation.

Also summarized functions in biological pathways which is very important and necessary for further research

In conclusion, this review marks a substantial stride in comprehending the genetic and molecular underpinnings of GH3 genes and their stress responsive functions in plants which might be useful for further plant stress resistance research and crop improvement.

 Many thanks to the reviewers for recognizing our manuscript.

Minor comments:

L179: adventitious root (AR) came only one time. So, no need AR

Similarly, Electrophoretic mobility shift assays (EMSA), and cytokinin (CK) and other if any

Some genes are not italicized. Check throughout the text

Done as suggested. We have carefully read through the manuscript and checked/corrected these unnecessary abbreviations.

L311-320: need some references if available

Done as suggested. Now these sentences were described as follows:The differential effects of Group II GH3 genes on pathogen stress between rice and Arabidopsis could be due to multiple factors. In rice, OsGH3s appear to primarily contribute to forming a robust cell wall barrier, which acts as the first line of defense against pathogen entry, without further invoking a hypersensitive response or SAR [38, 93]. In contrast, the expression of AtGH3 in Arabidopsis is highly specific to tissue type and developmental stage, and unspecific overexpression could lead to unintended consequences, such as disruptions in the levels of IAA and salicylic acid (SA), which are critical for the plant's defense response [90, 91]. Additionally, the GH3-catalyzed IAA-Asp conjugate is hypothesized to act as a susceptibility signal [22]. While GH3 activity reduces free IAA levels, it simultaneously results in the accumulation of IAA-Asp. The specific distribution and balance of IAA and IAA-Asp within the plant may significantly influence the outcome of GH3's role in pathogen response [91].’

Genes (transcription factors) names should be italic.

Done as suggested.

Some references were old, revised it accordingly where necessary.

We do our best to replace old references with the latest.

Some grammatical mistakes are present in the MS, revise the MS from the English speaker in the revised version.

We have carefully considered your comments, and have tried our best to revise any grammatical errors in the new manuscript. We also invited a native English speaker to revise the manuscript carefully.

Reviewer 2 Report

Comments and Suggestions for Authors

The authors have written an excellent comprehensive review showing the multifaceted role of Gretchen Hagen3 in plant species. They have beautifully compiled data from the literature and have included nice illustrations that help understand the functioning of the GH3 class of genes. 

1. I would suggest improving the first section which relates to the structural and functional specificity of GH3 enzymes. It would be fascinating to illustrate the different active sites of the enzyme and their different substrates.

2. Although the authors have discussed the multifaced role of GH3 in different responses some sections lack critical introspection. One such section is where it is stated that catalytic active sites of GH3 alter the substrate specificity. Why is this so? Can it indicate precursor/modified forms of GH3 in plants?

Author Response

The authors have written an excellent comprehensive review showing the multifaceted role of Gretchen Hagen3 in plant species. They have beautifully compiled data from the literature and have included nice illustrations that help understand the functioning of the GH3 class of genes. 

  1. I would suggest improving the first section which relates to the structural and functional specificity of GH3 enzymes. It would be fascinating to illustrate the different active sites of the enzyme and their different substrates.

 Done as suggest. We have added more about the active site and substrate selection of the enzyme, which are described as:Similarly, Leu137-Ser mutation leads to a benzoate/SA preference, and the replacement of Leu137 with Arg/Ile induces a preference for JA [24, 25]. ... Ser341 Participates in adenylate formation by forming hydrogen bonds with phosphate groups…Additionally, seven residues identified through sequence comparison of acyl-substrate binding sites (Arg130, Leu137, Valine 174 (Val174), Leu175, Methionine 337 (Met337), Alanine 339 (Ala339), and Tyrosine 344 (Tyr344)) are thought to be involved in the substrate-specific selection for IAA.

  1. Although the authors have discussed the multifaced role of GH3 in different responses some sections lack critical introspection. One such section is where it is stated that catalytic active sites of GH3 alter the substrate specificity. Why is this so? Can it indicate precursor/modified forms of GH3 in plants?

Thanks for your suggestion, we have added more discussion, which described as: ‘It has been observed that the residues involved in the AMP-binding site of acyl adenylate cleavage enzymes exhibit a high degree of conservation across the GH3 protein family. In contrast, the residues that interact with acyl substrates show variability, accommodating the binding of diverse substrates. [25, 26, 28, 29]. This variation in amino acid residues leads to the formation of different binding pockets, each tailored for specific substrates. [29]. This may account for the fact that auxin function at all stages of plant growth and development, a process that offers great flexibility in regulating auxin action is necessary. The existence of a complex auxin conjugation system, as evidenced by these variations in GH3 proteins, is likely a strategic evolutionary development to facilitate this flexibility. [28].

Reviewer 3 Report

Comments and Suggestions for Authors

This manuscript did a nice job reviewing the recent research findings on GH3s and its IAA conjugation related roles in plant growth and development processes. I have two minor suggestions:

1. In Introduction line 34, GH3 needs to be in full name.  

2. Authors discussed the atypical roles of GH3 and IAA-aa in section 5 which is the Concluding Remarks section. I feel that this part can be taken out and put into a separate section, as the Concluding Remarks usually are short and mainly conclusive, rather than discussing specific points.

Author Response

This manuscript did a nice job reviewing the recent research findings on GH3s and its IAA conjugation related roles in plant growth and development processes. I have two minor suggestions:

1. In Introduction line 34, GH3 needs to be in full name.  

Done as suggested.

2. Authors discussed the atypical roles of GH3 and IAA-aa in section 5 which is the Concluding Remarks section. I feel that this part can be taken out and put into a separate section, as the Concluding Remarks usually are short and mainly conclusive, rather than discussing specific points.

We have followed this excellent advice, and have rewritten the sentence to now say:

5. Atypical Roles of Group II GH3 and IAA-amino Acids

6. Concluding Remarks